# Zein-Based Films Containing Monolaurin/Eugenol or Essential Oils with Potential for Bioactive Packaging Application

**DOI:** 10.3390/ijms23010384

**Published:** 2021-12-29

**Authors:** Jana Sedlarikova, Magda Janalikova, Petra Peer, Lucie Pavlatkova, Antonin Minarik, Pavel Pleva

**Affiliations:** 1Department of Fat, Surfactant and Cosmetics Technology, Faculty of Technology, Tomas Bata University in Zlin, Vavreckova 275, 760 01 Zlin, Czech Republic; 2Department of Environmental Protection Engineering, Faculty of Technology, Tomas Bata University in Zlin, Vavreckova 275, 760 01 Zlin, Czech Republic; peer@utb.cz (P.P.); l_pavlatkova@utb.cz (L.P.); ppleva@utb.cz (P.P.); 3Department of Physics and Materials Engineering, Faculty of Technology, Tomas Bata University in Zlin, Vavreckova 275, 760 01 Zlin, Czech Republic; minarik@utb.cz; 4Centre of Polymer Systems, Tomas Bata University in Zlin, Trida Tomase Bati 5678, 760 01 Zlin, Czech Republic

**Keywords:** zein, film, essential oils, eugenol, monoglyceride, wettability, antibacterial activity, mechanical properties

## Abstract

Zein is renewable plant protein with valuable film-forming properties that can be used as a packaging material. It is known that the addition of natural cross-linkers can enhance a film’s tensile properties. In this study, we aimed to prepare antimicrobial zein-based films enriched with monolaurin, eugenol, oregano, and thyme essential oil. Films were prepared using the solvent casting technique from ethanol solution. Their physicochemical properties were investigated using structural, morphological, and thermal techniques. Polar and dispersive components were analyzed using two models to evaluate the effects on the surface free energy values. The antimicrobial activity was proven using a disk diffusion method and the suppression of bacterial growth was confirmed via a growth kinetics study with the Gompertz function. The films’ morphological characteristics led to systems with uniform distribution of essential oils or eugenol droplets combined with a flat-plated structure of monolaurin. A unique combination of polyphenolic eugenol and amphiphilic monoglyceride provided highly stretchable films with enhanced barrier properties and efficiency against Gram-positive and Gram-negative bacteria, yeasts, and molds. The prepared zein-based films with tunable surface properties represent an alternative to non-renewable resources with a potential application as active packaging materials.

## 1. Introduction

In recent years, the study of films based on renewable resources has acquired increased attention. Zein, as a renewable plant protein, is a suitable candidate for the preparation of safe and ecologically favorable polymer carriers of bioactive compounds. It is known for its valuable film-forming properties that predetermine its applications as packaging materials [1]. Since protein films can suffer from worse mechanical and barrier properties, some previous studies have focused on overcoming these problems. Khalil et al. [1] investigated the effects of natural cross-linkers, such as citric acid, succinic anhydride, and eugenol, on the physical characteristics of solvent-cast zein-based films. An enhancement of the tensile properties was revealed due to the modification of mentioned cross-linking agents.

The addition of various bioactive compounds in a polymer matrix has also been intensively studied to produce materials to prevent the colonization and growth of various microorganisms. Essential oils or their components can be considered effective natural compounds that provide antimicrobial and antioxidant properties. Different essential oils were encapsulated into zein-chitosan films in the study by Escamilla-García et al. [2], where the modification resulted in decreased water vapor permeability and improved mechanical properties. Eugenol is a natural phenolic compound occurring in clove and cinnamon essential oils. Due to its antioxidant and antimicrobial activity, it has been applied in the cosmetic, pharmaceutical, and food industries [3]. Partial esters of glycerol and medium-chain fatty acids (such as monocaprin and monolaurin) are other candidate antimicrobial agents that can be applied to fight microbial contamination in foods, cosmetics, and pharmaceutical products. These compounds have amphiphilic structures; thus, they contain both hydrophilic and hydrophobic parts within their molecules [4]. Monoglycerides of capric and lauric acid are known for their antimicrobial properties, which interfere with bacterial cells, leading to the inhibition of the cell metabolism or cell lysis [5,6].

Incorporating the mentioned active compounds into polymer films can affect their physicochemical properties and performance in the final application. The aim of this study was to investigate zein films enriched by fatty acid esters, monoglycerides of lauric acid, and their combinations with other natural bioactive phenolic compounds and to report on the potential synergic activity against microorganisms. The main aim was to find and evaluate a suitable formulation to obtain films with good mechanical and barrier properties with additive value in the form of enhanced antimicrobial activity. The potential to control the surface and mechanical properties of the developed films via the incorporation of active agents was revealed. The resultant systems based on biodegradable and renewable materials and prepared using eco-friendly techniques could significantly contribute to achieving a circular economy in packaging technologies.

## 2. Results

Two sets of test samples were prepared and compared with a pure zein film (control). The first set of films contained two-component films, containing zein and one active agent (eugenol (EU), thyme (TH), oregano (O), and monolaurin (ML)). The second set of samples contained three-component films, containing zein/ML and one of the above-mentioned active agents. The active compounds were mixed in various ratios (see Table 1). The prepared non-transparent films were of yellow shades, depending on the specific formulation.

### 2.1. Mechanical Properties

Zein films commonly suffer from poor mechanical characteristics with higher brittleness. To overcome this limitation, different additives and plasticizers have been studied to improve the flexibility of these protein-based films [7]. The prepared samples were analyzed using the tensile and puncture tests, since these aspects play a crucial role in determining the integrity of packaging materials during the application, transport, and storage processes. The highest tensile strength, which is defined as the maximum stress exerted on a sample [8], was shown by the control zein sample (11.6 MPa), as can be seen in Figure 1a. Following the incorporation of active compounds, a drop in tensile strength was recorded, with the values ranging from 1.5 MPa for the 5ML/2O zein film to 7.9 MPa for the film containing 3 wt% of oregano oil. Decrease tensile properties caused by the addition of various hydrophobic agents into the polymer matrix was also seen in other studies [8,9]. The reason for this may be the loosened polymer structure due to the mutual estrangement of polymer chains leading to the decreased tensile characteristics. An opposite trend was observed in the study by Vahedikia et al. [10], where the encapsulation of cinnamon essential oil resulted in increased tensile strength and simultaneously decreased ductility.

The percentage of elongation at fracture indicates the degree of material elasticity. The statistically significant (*p* < 0.05) differences among the individual samples are shown in Figure 1b. In contrast, the control zein film exhibited a relatively low value (12%), while the elasticity of the films enriched with eugenol was increased dramatically by almost 97%. Even the samples with combinations of monolaurin and eugenol and monolaurin and oregano exhibited high elongation values. Eugenol’s plasticizing effects (297% elongation at break in the films with 1% eugenol) were also shown in the study by Boyaci et al. [11]. This phenomenon may be the consequence of H-bonds between –OH groups of phenolic compounds and peptide carbonyl groups of zein protein. On the other hand, a different trend was revealed in the samples, including those containing only monolaurin or individual essential oils. In this case, the values did not exceed the elongation of the control zein film. The usual trend was confirmed that a less rigid structure with lower tensile strength was associated with increased material elasticity and vice versa, which as shown by comparing Figure 1a,b [12]. The lower plasticizing activity of thyme oil could be ascribed to phenolic–phenolic interactions between thymol molecules, which are predominant in this type of essential oil [8,12].

Similarly, in the work by Pereira et al. [13], the drop in elongation values was monitored after the incorporation of thyme and garlic essential oils into the zein matrix. This effect could be the consequence of porous structure formation in the polymer film. It can be stated that while incorporating some types of EO may reduce the mechanical properties, other types or the combination of active compounds can lead to a significant increase. Thus, the results are specific to the selected concrete combination of polymer base and bioactive molecules, which can be difficult to generalize and extrapolate to other systems [14]. The synergic effect between monolaurin and eugenol can be confirmed from the obtained data. Eugenol here evidently promotes the elasticity of the samples, imitating the function of plasticizers. Its positive effect was even proven in the study by Nining et al. [15], who compared the activity of this substance with that of oleic acid in dextromethorphan hydrobromide films. It is known that plasticizers can affect the processing properties due to the lower viscosity, easier dispersion, and lower processing temperatures and energy inputs [16].

A puncture test was applied to analyze the effect of the modification on the sample integrity during the penetration [8]. The modification of zein with active molecules led to a significant (*p* < 0.05) decrease in puncture strength, with the lowest value being for the Z/3O sample of 8 N/mm (Figure 1c). A similar trend, i.e., a decrease in puncture strength, was observed in the chitosan samples modified with the thyme essential oil [8]. The values of puncture deformation (Figure 1d) showed an increase in most modified samples were compared to the reference zein film without any bioactive compound. The most significant change (46%) was shown in the film containing 3 wt% monolaurin and 2 wt% thyme oil. The essential oils of oregano and thyme are complex mixtures of components that differ according to the locality, climatic conditions, and time of plant collection. Although both selected EOs could be grouped into the mixed composition type [17], they include different contents of individual components. Based on the previously carried out gas chromatography analysis [8,18], the dominant components are thymol and p-cymene and carvacrol, p-cymene, and γ-terpinene for thyme and oregano oil, respectively. Presumably, this is the reason for the different data obtained for samples Z/3T and Z/3O in the puncture tests, where the difference of almost 65% was obtained during the puncture strength measurement (Figure 1c).

### 2.2. Thickness of the Films

The thickness is considered a significant characteristic affecting other properties of a polymer film, especially the permeability for water and gases. The thickness of the control zein film was 258 µm. Adding the essential oils or eugenol increased the thickness, as the film with eugenol content gave the highest value of 284 µm. This trend was observed in the study by Sedlarikova et al. [8], where the modification of chitosan films by thyme essential oil was studied. On the other hand, the modification of zein with monolaurin led to decreased thickness (the lowest value was revealed in the Z/3ML sample), as shown in Table 2. Monolaurin belongs among the nonionic surface-active agents used to form dispersion systems and stabilize them. The decrease in thickness after monolaurin addition into zein films could be the consequence of the different organization of this amphiphilic agent in the zein matrix (as proved by SEM analysis).

### 2.3. Water Vapor Permeability

The barrier properties of polymer films and packaging are considered a crucial factor in preventing or minimizing moisture transfer between the product and the surrounding atmosphere. In order to prolong the shelf life and heighten the quality of packaged products, the water vapor permeability should be as low as possible [19].

Data from WVP measurements are shown in Table 2. A significant drop in water permeability was seen in modified films compared to the control zein sample. The greatest decreases (compared to the average of 1.3 g/Pa.h.m^2^) were seen in the films containing individual essential oils, oregano, or thyme. This phenomenon could result from the mutual interactions between the zein network and the included phenolic compounds, making the hydrogen groups used for forming hydrophilic bonds with water less available. A similar trend was observed in the study by Ghasemlou et al. [19], who investigated corn starch films incorporated with plant essential oils from Zataria multiflora Boiss and Mentha pulegium. Monolaurin incorporation resulted in higher WVP values, probably due to hydrophilic groups oriented out of the film surface. These results correlate with the measured thicknesses of the zein films.

Moisture barrier properties were also improved in the study by Moradi et al. [20], whereby the chitosan films were enriched with Zataria multiflora essential oil, which caused a decrease in the transmission rate in comparison to pure chitosan films. Besides the bioactive molecules, plasticizers can affect the barrier properties of polymer films. Higher water vapor transmission values were induced by the higher concentrations of glycerol and sorbitol plasticizers incorporated in whey protein films [21]. The hydroxyl groups of the plasticizers can change the polymer–polymer interactions, leading to the development of the polymer–plasticizer hydrogen bonds and increasing the intermolecular spacing and transmission of the polymer film [22].

### 2.4. Surface Properties

The surface properties, indicating the material hydrophobicity or hydrophilicity of the polymer films, mainly affect the final applications of these polymeric materials. The wettability of the films was analyzed via the contact angle measurements, defined as the angle between the tangent conducted from the contact site of the liquid droplet with the film’s surface. The contact angle for the water in the control zein film was 56°, similar to the study of Luís et al. [7], who investigated solvent-cast zein films enriched with glycerol and licorice essential oil. The surface wettability is also affected by the solvents used for the film preparation. Shi et al. [23], at a particular solvent concentration, obtained more hydrophobic films cast from acetone/water mixture (about 75°) when compared to an ethanol/water (about 40°) solvent. Higher hydrophobicity, with a contact angle over 74°, was also shown in [24]. The difference in the measured data may have been caused by the presence of the plasticizer. Moreover, the properties of the test liquids and the overall character of the measured substrate, such as homogeneity and roughness, play important roles in the final material wettability [25].

It can be observed from Figure 2 that the contact angles for water differed depending on the type of active compound or their combinations included in the zein matrix. For the individual phenolic active substances added (oregano, thyme, and eugenol), slight decreases in values were observed compared to the control zein film. On the other hand, significant increases in hydrophobicity were observed in the films enriched with monolaurin and its combinations with essential oils. When the monolaurin content was increased from 3 to 5 wt%, the contact angle decrease was proven, except for the sample combined with oregano oil. A similar trend was observed in the study on polyvinyl-butyral-based membranes enriched with monoglyceride of capric acid [26].

Surfaces can be classified into two groups, i.e., hydrophilic (wetted by water, high surface energy) and hydrophobic (non-wetted by water, low surface energy) [27]. All samples showed contact angles lower than 90°, indicating wettable surfaces.

The surface energy values evaluated using the OWRK model ranged from 45 to 83 mJ/m^2^ and are summarized in Table 3. Modifications by the selected active compounds caused increased values in comparison with the control zein film, indicating higher wettability. On the other hand, the films containing monolaurin and its combinations with oregano and thyme essential oil showed lower values of γ_tot_, in accordance with the acquired higher contact angle values (Figure 2).

The total surface energy γ_tot_ is further divided into the dispersion γ_d_ and polar γ_p_ parts. The obtained data showed that the latter predominates in almost all tested zein samples, regardless of the type and concentration of added active compound. The most significant change (61%) in surface energy was observed in the Z/3ML sample as compared to the reference. The highest polar components values were revealed in the films with the monolaurin/eugenol combination. Presumably the hydrophobic part of the amphiphilic ML molecule was adsorbed toward the polymer matrix, whereas the polar part was oriented into the environment. A similar trend can be seen in Table 4, showing the surface energies calculated using the Wu model. The lower total surface energy values (around 20 mJ/m^2^) were calculated for the sample with 3 wt% monolaurin and the films containing the combinations of monolaurin and essential oils of oregano and thyme, similarly to the previous model. The wettability of surfaces also depends on the material of the plate or dish used for solvent casting, as was proven by Malm et al. [28]. It was confirmed that the zein solution mimics the characteristics of such materials due to the possibility of interacting with different parts of its helical structure. The hydrophilicity of their zein-based films was significantly higher in samples cast on PDMS compared to PS and PTFE. Consequently, the calculated surface energies, specifically their polar and dispersive parts, differed according to the mold used for the casting process.

### 2.5. FTIR Analysis

Mutual interactions between the zein polymer and selected active compounds were analyzed via FTIR-ATR spectroscopy (Figure 3). The spectrum of the control zein film (Figure 3a) shows the peaks characteristic for this type of polymer corresponding to amides I, II, and III at 3290 cm^−1^ (-NH_2_), 1645 cm^−1^ (C=O), and 1538 cm^−1^ (N-H, C-N), respectively. The peak at 1645 cm^−1^ confirms the presence of the higher amount of the α-helical secondary structure. On the other hand, the β-structure cannot be seen in the obtained spectra due to the absence of peaks at 1614 cm^−1^ and 1631 cm^−1^ [29]. The peaks at 1729, 2849, and 2957 cm^−1^ can be observed in the zein film enriched with active agents (Figure 3b,c), revealing the C=O and C-O stretching pertaining to the structure of the monolaurin molecule. A predominant role of monolaurin was confirmed, since the spectra of the sample with monolaurin and thyme essential oil did not reveal any significant changes [30].

### 2.6. Differential Scanning Calorimetry (DSC)

Differential scanning calorimetry was used to study the potential interactions between the zein polymer and active compounds. The results from the first heating scan shown in the DSC pattern, for which the temperatures ranged from 0 to 150 °C, are shown in Figure 4. The thermogram of monolaurin showed a sharp endothermic peak at 65 °C, corresponding to the melting temperature of this surfactant. A comparable result was obtained by Chinatangul et al. [31]. The zein samples with monolaurin and its combinations with polyphenolic compounds exhibited suppressed endothermic peaks at a similar position. The glass transition temperature was not observed in these formulations, possibly as the consequence of the overlapped Tg of zein and the melting event of monolaurin [32]. The film containing 3 wt% thyme oil demonstrated a different result due to the absence of monolaurin in the mixture. A small endothermic peak of about 74 °C was shown in this thermogram. In the unmodified zein film, a peak at 67 °C was observed, corresponding to the glass transition temperature. A similar Tg value (68 °C) for the zein film was achieved in the study by Bueno [33]. The shift of this endothermic peak in the zein film toward zein modified with thyme oil could indicate the partial miscibility of these two components [34]. Generally, the presence of only one transition temperature is regarded as proof of the miscibility of individual components [32]. On the other hand, two peaks at 54 and 66 °C were revealed in the sample containing only 3 wt% monolaurin, which could indicate the worse miscibility of the zein polymer and monoglyceride, as was also supported by SEM analysis (Figure 5).

### 2.7. SEM Analysis

The effect of zein modification with active agents was observed by scanning electron microscopy (SEM) (see Figure 5). The control zein films exhibited a relatively compact homogeneous structure with fine pores (196 ± 34 nm). The smaller cracks were probably caused by the relatively fragile zein films structure. To a greater extent, a similar phenomenon was observed in the study by Pereira et al. [13]. Two phases were shown in the surface morphology of the zein films [11], indicating that the phase separation process occurred between the polymer and plasticizer (glycerol).

Figure 4 shows the SEM photographs of surfaces and cross-sections of zein films modified with essential oils, eugenol, and monolaurin. The significant morphological changes in the pores (with diameter increases of 334 ± 135, 501 ± 181, and 958 ± 440 nm on the surface and 1794 ± 492, 1 945 ± 623, and 4905 ± 1 970 in the cross-section for Z/3O, Z/3T, and Z/3EU, respectively) dispersed uniformly in the polymer matrix were proven in comparison with the control film. However, the surface of Z/3ML film exhibited an entirely different structure—a non-porous surface with flat plates. The addition of monolaurin caused the formation of irregular “flat sheet plate” structures caused by the polymorphic character of fatty acid monoglycerides.

The films containing the combination of active compounds, essential oils, eugenol, and monolaurin are shown in Appendix A. In comparison with the single-component samples, a significant difference in the film structure was visible. The mentioned flat plate structures were accumulated mainly in the upper part of the film, as can be seen in the film cross-section. On the other hand, the droplets that arose due to the essential oil content were present in the polymer matrix volume. These morphological changes affected the mechanical properties (as discussed in Section 2.1.), especially the tensile strength, as compared to reference zein film. The higher monolaurin content decreased the tensile strength and the increased film elasticity (see Figure 1a,b).

### 2.8. AFM Analysis

The surface topographies of the zein films with active agents were observed by atomic force microscopy (AFM) (see Figure 6). The control zein film exhibited a relatively flat surface with a surface roughness Sa = 18 nm and maximal height difference Sz = 0.84 µm. A similar surface roughness value (Sa = 19 nm) was obtained by Malm et al. [28]. The films modified with active compounds showed Sa values ranging from 61 nm for Z/3T to 399 nm for Z/3ML. It is obvious that samples containing monolaurin showed increased surface roughness, which is in agreement with the SEM results (Figure 5). The correlation between the surface roughness (Figure 6) and the measured contact angles can be observed in Figure 2. The addition of monolaurin resulted in higher water contact angle values. On the other hand, the sample with eugenol showed higher wettability. Based on these results, a dominant effect of the added active compounds was presumed. Nevertheless, the contribution of the surface roughness to the surface wetting properties cannot be fully neglected here, similarly as discussed in our previous work dealing with the treatment of polystyrene surfaces [35,36,37,38]. It is evident that more factors have to be considered, as follows from theoretical models (e.g., Wenzel and Cassie–Baxter models) describing the influence of the structure and chemical composition of the surface on wetting [37,38,39,40].

### 2.9. Antimicrobial Properties

The antimicrobial properties were analyzed using the agar disk diffusion method against the selected microorganisms from the Bacteria (*E. coli*, *S. aureus*) and Eukarya (*C. albicans*, *A. niger*) domains. The inhibition zones of the tested samples are summarized in Table 5 and Table 6.

It was observed that the control zein film did not exhibit any inhibition effect, regardless of the tested microorganism, similarly to the study by Kashiri et al. [41]. Even zein films with thyme were not active against the tested microorganisms. On the other hand, a significant effect was observed against the Gram-positive bacterium *S. aureus* when a bioactive compound of oregano, monolaurin, or eugenol was added. All films with eugenol were active, even against the Gram-negative bacterium *E. coli*. Eugenol’s activity in zein films was also proven in the study of Boyaci et al. [11], who investigated its effects against the pathogenic bacteria *Listeria innocua* and *Escherichia coli*. A lower antibacterial activity against *E. coli* could be caused by a different character of the bacterial cell wall. It is known that while a Gram-positive cell wall possesses mainly peptidoglycan with a small number of proteins, a cell wall of Gram-negative bacteria is more complex with a stable lipopolysaccharide layer, increasing the resistance [10].

By comparing all samples, it can be confirmed that monolaurin addition significantly increased the antimicrobial activity (even of thyme oil), mainly against Gram-positive *Staphylococcus aureus*. Moreover, enhanced inhibition of the samples with higher eugenol content against Gram-negative *E. coli* was confirmed.

The agar diffusion test against yeasts and molds showed no effect of the control zein films, similarly to bacteria. No inhibition zone was proven against *Aspergillus niger*, even in the sample with only eugenol (without monolaurin). Although the higher monolaurin concentration was not directly proportional to greater effect, the positive trend in the samples enriched with the active compound combination was proven (Table 6). The advantage of the further application of monolaurin’s antifungal activity is pH independence [42]. The antifungal activity levels of eugenol and monolaurin against fungi of the *Aspergillus* genera were assessed in an earlier study. The study showed that monolaurin was less effective than eugenol; nevertheless, their combination was not studied. On the other hand, this effect can also inhibit aflatoxin production [43].

The antibacterial effects of the prepared films were studied in more detail via the growth kinetics. Over time, bacterial proliferation can be routinely determined by measuring the optical density (OD_600nm_) and reliably investigating the antimicrobial activity [44]. The initial lag phase consists of limited growth as bacteria adapt to their new environment. The highest rate of bacterial growth defines the exponential phase. The third stationary phase is characterized by the equilibrium containing dividing and dying cells as nutrients are depleted and waste products accumulate. In this study, the OD_600nm_ measurement was performed on a microplate with the bacterial inoculum itself or in the presence of zein or modified zein films. The Gompertz model has a sigmoid shape with a clear inflexion point [45].

Population growth kinetics concisely described by growth parameters (λ, μ_max_, and A) were calculated using this model (Table 7). The bacterial growth curves in the presence of zein were not significantly different to those measured without. On the other hand, the addition of a modified film (Z/3O, Z/5ML/2O, Z/3EU, Z/5ML/2EU) to the test well caused total growth inhibition of *E. coli* and *S. aureus*. Thus, all tested compounds or their combinations at the presented concentration (oregano, eugenol, or monolaurin) exhibited antibacterial activity, as stated in the literature [3,5,46], which is the proof of their release from the films.

## 3. Discussion

The modification of the zein polymer with various polyphenolic compounds and a nonionic monoglyceride surfactant led to changes in the physicochemical and antimicrobial properties. It is well known that pure zein films suffer from worse mechanical characteristics since they are too brittle and stiff, limiting their technological processing and application. To improve the mechanical properties, various modifications of zein films have been investigated, e.g., combinations with other polymers such as chitosan or polyvinyl alcohol [33], using a mixture of glycerol and oleic acid [47], or introducing different types of nanoparticles [48].

The mixture of cinnamon essential oil and chitosan nanoparticles used in the study [10] led to a mild increase in tensile strength. Despite this fact, the tensile properties of zein films enriched with active agents prepared in our study exceeded their results, especially for the Z/3O and Z/3T films (that achieved the tensile strength of 8 and 7 MPa, respectively). Even in samples with monolaurin and its combination with 2 wt% of eugenol and thyme oil, the tensile strength was higher (average 3.7 MPa) than in the referenced study, which was about 2 MPa [10]. TiO_2_ nanoparticles were incorporated to enhance the mechanical properties of zein/chitosan films in the study by Qu et al. [48]. The maximum tensile strength (28 MPa) was reached at 0.15 wt% TiO_2_, while higher content caused a decrease in the mechanical strength. This phenomenon may have been due to the weakened regularity of the polymer network structure and the potential aggregation of nanoparticles at higher concentrations. Composite films based on zein and methyl cellulose were investigated in the study by Zhou et al. [49]. The addition of the cellulose derivative significantly enhanced the tensile characteristics compared to pure zein films. It is evident that the specific components of the added essential oils, as well as their mutual interactions with the polymer matrix, play an important role in the resultant physical properties of the films.

A significant change was observed in the elongation parameters in the modified zein films, especially with eugenol, where the level of elasticity increased dramatically (to more than 300%), making it a potential multifunctional bioactive substance, both for antimicrobial and material characteristics. In general, the higher flexibility of polymer films is associated with stronger interactions between individual components—in this case between carbonyl groups of the zein polymer and H-bonds between hydroxyl groups of eugenol with a polyphenolic structure.

Weaker interactions between individual components were presumed from the obtained results, even though thymol and carvacrol, as the main constituents of used EOs, exhibit similar molecular structures (see Figure 7). The difference is in the location of –OH groups in the molecules, which can cause their lower availability for interaction with zein polymer. A similar phenomenon may have hypothetically occurred in the zein sample with only monolaurin, with the –OH group at the sn-1 position esterified in the structure (Figure 7).

Various types of plasticizers have been studied to improve the elasticity of zein film. The effect of polyethylene glycol (PEG) as potential plasticizer of zein/methyl cellulose films was studied by Zhou et al. [49]. While the PEG plasticization effect was poor in pure zein films due to their low compatibility, the same plasticizer caused a more significant increase in elongation in composite films based on zein and methyl cellulose. Incorporating nanocarbonate in the concentrations from 1 to 3 wt% ensured the higher flexibility of zein oleic acid films [12].

The effects of plasticizers depend on several factors, such as the chemical structure, ratio of polar and non-polar groups, molecular shape (linear or branched), and molecular weight. These aspects affect the miscibility of individual components and final homogeneity [50]. Generally, the mutual compatibility between the base polymer and added components, including the plasticizers, is controlled in biopolymer-based films and layers. As evidenced by SEM analysis (Figure 5), no phase separation was observed in the zein films modified with active agents or combinations. Moreover, substances with lower molecular weight support higher plasticizing activity, which could be the reason for obtaining the zein/eugenol films with higher levels of elasticity than with the essential oils (thyme, oregano). Eugenol has a dual function, with both antimicrobial and plasticization actions, which could help in developing high-performance materials.

The wettability testing proved that the surface properties of the developed zein films could be tuned using the selected active agent or a combination. In two-component films containing only oregano, thyme, or eugenol, slight decreases in contact angles were revealed compared to the unmodified zein film, with significantly higher contact angle values being recorded in the samples containing ML or the combinations with EOs. A correlation between the measured contact angles and the surface roughness was observed. The addition of a bioactive compound to the polymer matrix can efficiently control the surface hydrophilicity. In the study by Shi et al. [23], the contact angles of zein-based films significantly decreased to values lower than 10° over a very short timeframe (2 min) due to the combination of UV light and ozone treatment. The samples with plasticizers can exhibit hydrophilicity increases followed by potential hygroscopic effects. For this reason, the contact angle values can significantly change even during storage. In the samples with glycerol, the increment of the contact angle from 58° to 61° was proven over one week, which is explained by the plasticizer migration and partial moisture loss [51]. The contact angle measurement and surface energy evaluation showed that relatively low concentrations of selected bioactive compounds could significantly modify the surface hydrophilicity.

The SEM analysis demonstrated significant morphological changes in the modified zein films compared to the control film. Single-component samples with oregano, thyme oil, and eugenol showed pores of variable size dispersed uniformly in the polymer matrix. The sample with 3 wt% monolaurin exhibited a non-porous structure with flat plates due to the polymorphic character of the monoglycerides.

The results of the thermal analysis using DSC showed the effects of modification of the zein with active compounds on the glass transition temperature, proving the partial miscibility of the components. This could be the reason for the lower inhibition activity against the Gram-negative bacteria *E. coli* and yeasts and molds, because of the hindered diffusion from the sample. On the contrary, no significant effect on the T_g_ value was demonstrated in the study by Ordon et al. [52], who modified polyethylene films with a mixture of plant extracts. On the other hand, good antimicrobial properties were observed during the agar diffusion test against Gram-positive *S. aureus*. In addition, the activity against *E. coli* and *S. aureus* was proven during testing of the growth kinetics. Moreover, the synergic effect between monolaurin and polyphenolic compounds was proven. Interactions between eugenol and monolaurin were investigated by Blaszyk [53] against common meat spoilage and pathogenic microorganisms, and synergic effects were proven. However, to our knowledge, the incorporation of such a combination into a polymer matrix has not been reported yet.

## 4. Materials and Methods

### 4.1. Materials, Chemicals and Microorganisms

Zein polymer (total nitrogen 13.0 to 16.0%) was obtained from TCI (Deutschland GmbH). Monoglyceride of lauric acid was prepared according to the procedure reported in [8]. Eugenol was purchased from Sigma Aldrich (Prague, Czech Republic). Essential oils of oregano and thyme were supplied by Nobilis Tilia (Krasna Lipa, Czech Republic). The chemical structures of the investigated active compounds are shown in Figure 7. Microbial strains, i.e., *Escherichia coli* ATCC 25922, *Staphylococcus aureus* ATCC 25923, *Candida albicans* CCM 8215, and *Aspergillus niger* CCM 8155, were obtained from the Czech Collection of Microorganisms (CCM, Czech Republic).

### 4.2. Preparation of Zein Films

Zein-based films were prepared following the process used in [54] with slight modifications, as schematically displayed in Figure 8. The mixture of zein powder and ethanol (17 wt%) was homogenized under continuous stirring (30 min), after which the filtration was implemented to remove undissolved particles. Next, glycerol was added (4 wt%) and this mixture was boiled for 5 min. When the solution cooled down, the bioactive agents were added in defined concentrations (see Table 1). The solution without active substances was used as a control.

The prepared mixture (10 mL) was cast into Petri dishes (9 mm in diameter) and left to dry in an air-circulated oven (30 °C, 24 h). Before further testing, the samples were stored at 25 °C and 60% relative humidity). The appearance of the prepared films after solvent casting and drying is shown in Appendix A.

### 4.3. Mechanical Properties

Mechanical properties were characterized using TA1 Series texture analysis equipment (AMETEK Test and Calibration Instruments, Largo, FL, USA). The evaluation was performed using NexygenPlus texture analysis software (version 4.0.1.184). The samples were stored at 25 °C and 60% relative humidity for 48 h. The tensile test proceeded at a speed of 1 mm.s^−1^ on the prepared samples (1 × 6 cm) mounted into the tensile grips (the exposed area of specimen was 1 × 4 cm). Tensile strength (TS) was calculated according to Equation (1):(1)$$\mathrm{TS}=\frac{F_{t}}{TW}$$
where *F_t_* is the maximum tensile strength (N), *T* is the average thickness of the film sample (mm), and *W* is the width of the film sample (mm).

The elongation at break (*E*) was determined as a percentage by dividing the elongation at the break point by the initial specimen length multiplied by 100. The resultant values were calculated from four measurements.

Before the puncture test, the square samples measuring 4 × 4 cm^2^ were cut and placed between the plates with a circular hole measuring 2 cm in diameter. After securing a sample with clamps, a cylindrical probe measuring 2 mm in diameter was pressed through the sample (speed 1 mm.s^−1^). The puncture strength (*PS*) calculation in N.mm-1 was performed according to Equation (2):(2)$$PS=\frac{F_{max}}{T}$$
where *F_max_* is the maximum puncture strength (N) and *T* is the average thickness of the film sample (mm). The puncture deformation (PD) in mm was evaluated from the distance when the probe contacted the specimen and the break point.

### 4.4. Thickness

The thickness of the films was gauged with a digital micrometer (Schut, Trossingen, Germany). The results are the averages of five measurements from each film at an accuracy level of ±0.006 mm.

### 4.5. Water Vapor Permeability

Barrier properties were analyzed by measuring the water vapor permeability (WVP) according to ASTM E96–95 [55]. The films were sealed at the mouth of a test dish filled with distilled water to reach 100% relative humidity, then placed in a desiccator containing silica gel to obtain 0% relative humidity. The dishes were periodically weighed until reaching a constant weight and the obtained values graphed vs. time were used for the calculation of the water vapor transmission rate (WVT) in g·h^−1^·m^−2^ and the water vapor permeability (WVP) in g·Pa^−1^·h^−1^·m^−2^ according to Equations (3) and (4):(3)$$\mathrm{WVT}=\frac{\left(\frac{m}{t}\right)}{A}$$
(4)$$\mathrm{WVP}=\frac{\mathrm{WVT}}{\Delta p}=\frac{\mathrm{WVT}}{S\left(R_{1}-R_{2}\right)}$$
where *m*/*t* is the weight loss vs. time (g·h^−1^), *A* is the test area in m^2^ (the mouth area of the test dish), Δ*p* is the difference in water vapor pressure (Pa), *S* is the saturation pressure of the vapor at the tested temperature, *R_1_* is the relative humidity in the test dish expressed as a fraction, and *R_2_* is the relative humidity at the vapor sink expressed as a fraction.

Data were expressed as means ± standard deviation (SD).

### 4.6. Surface Properties

The surface properties of zein samples were studied using the sessile droplet method by measuring the contact angle on an Attension Theta optical tensiometer (Biolin Scientific, Sweden) in combination with OneAttension software at ambient temperature. Owens, Wendt, Rabel, Kaelble, and Wu two-component models were used to evaluate the polar $\gamma^{p}$ and dispersive $\gamma^{d}$ components and to calculate the total surface energy. Distilled water and ethylene glycol were applied as the reference liquids, with the droplet volume equalling 5 µL at laboratory temperature. The final values were calculated as averages from five independent measurements. The OWRK geometric mean method and the Wu harmonic mean method are based on Equations (5) and (6), respectively [56]:(5)$$\sqrt{\gamma_{SG}^{d}\gamma_{LG}^{d}}+\sqrt{\gamma_{SG}^{p}\gamma_{LG}^{p}}=0.5\gamma_{l}\left(1+cos\theta \right)$$
(6)$$\left[\frac{\gamma_{SG}^{d}\gamma_{LG}^{d}}{\left(\gamma_{SG}^{d}+\gamma_{LG}^{d}\right)}+\frac{\gamma_{SG}^{p}\gamma_{LG}^{p}}{\left(\gamma_{SG}^{p}+\gamma_{LG}^{p}\right)}\right]=0.25\gamma_{l}\left(1+cos\theta \right)$$
where θ is the contact angle and $\gamma_{s}$ and $\gamma_{l}$ are the surface energies of the solid and liquid, respectively.

### 4.7. Fourier-Transform Infrared Spectroscopy (FTIR)

A spectroscopic analysis of the zein films was carried out on a Nikolet 6700 instrument (ThermoFisher Scientific, Waltham, MA, USA) in the spectral range of 400 to 4000 cm^−1^ with a resolution of 2 cm^−1^ (64 scans). An evaluation of the spectra was performed using OMNIC Paradigm software (Version 7.5).

### 4.8. Differential Scanning Calorimetry (DSC)

The thermal analysis was performed using differential scanning calorimetry (DSC) on a Mettler Toledo DSC 700/1 device (USA) placed in the temperature range of −50 °C to +300 °C at a heating rate of 10 °C/min under N_2_ atmosphere; the samples weighed approximately 6 mg. The results were evaluated from the first heating cycle of the samples.

### 4.9. SEM Analysis

Scanning electron microscopy (SEM, Tescan, VEGA 3) was used to study the zein films’ morphologies after gold layer coating via a sputter system. Adobe Creative Suite software was used for determination of the mean pore size calculated from 40 values.

### 4.10. Atomic force microscopy

Dried zein-based films were characterized with an AFM microscope using the ScanAsyst mode on a Dimension ICON instrument (Bruker Corporation; Billerica, MA, USA). Analyses were performed under laboratory temperature and humidity. A ScanAsyst-Air silicon nitride probe (Bruker Corporation; USA) with a spring constant of 0.4 N m^−1^ was used. Data were acquired on a scanning area of 50 × 50 μm. The scanning rate was 1 Hz. AFM data were processed in Gwyddion 2.5 software (Czech Metrology Institute, Jihlava, Czech Republic). The height profiles, maximal surface height (Sz), and surface roughness (Sa) were determined.

### 4.11. Antimicrobial Properties

The agar disk diffusion method was carried out with circular samples (9 mm in diameter) of control zein films and films enriched with bioactive compounds, which were placed on agar plates previously inoculated with 1 mL of 0.5 McF turbid microbial suspension (*Escherichia coli*, *Staphylococcus aureus*, *Candida albicans*, *Aspergillus niger*) in sterile saline solution. The plates with bacteria were incubated at 37 °C and with *Candida* and *Aspergillus* fungi at 30 °C/3–5 days. The whole experiment was performed in triplicate. The inhibition zones, as well as growth under the samples, were evaluated.

The growth kinetics of two bacterial species (*Escherichia coli*, *Staphylococcus aureus*) were studied to examine changes in the growth curve in the presence of the film disks (9 mm) containing antimicrobial compounds. The microplate wells were filled with 250 µL Mueller–Hinton broth (Himedia Laboratories Pvt. Ltd., Mumbai, India), 5 µL 0.5 McF turbid bacterial inoculum, and a sample disk with (monolaurin, thyme, oregano, or eugenol) or without (bacterial control growth) antimicrobial compounds (see Table 1). The microplate was incubated with shaking at 37 °C for 24 h and the values of absorbance (in nine repetitions) were read as optical density (OD_600nm_) values every 30 min using a Tecan microplate reader (M200Pro, Tecan, Männedorf, Switzerland). To evaluate the antimicrobial activity of zein films, the modified Gompertz equation was used to describe the lag phase of bacterial growth [57,58]. A non-linear regression analysis (Marquardt–Levenburgova method) was used for the calculation of the parameters μ_max_, λ, and *A* for the following conditions: μ > 0, λ > 0, and *A* > 0. The maximum specific growth rate (μmax) and asymptotic value were calculated as follows (Equation (7)):(7)$$y=A\cdot exp\left\{-exp\left[\frac{\mu_{max}\cdot e}{A}\left(\lambda -t\right)+1\right]\right\}$$
where μ_max_ is the maximum specific growth rate (log CFU.l^−1^.h^−1^), λ is the lag phase (h), and *A* is the asymptote, defined as the maximum value of relative microorganism counts (log CFU.l^−1^).

### 4.12. Statistical Analysis

All experiments were performed at least in triplicate. The obtained values were then calculated as means ± standard deviations using MS Office Excel software (Microsoft, 2020). The statistical evaluation of the experimental data was carried through one-way analysis of variance (ANOVA) using Statistica software (version 10, StatSoft, Inc., Tulsa, OK, USA) at the significance level of *p* < 0.05.

## 5. Conclusions

The main purpose of this study was to develop the optimum film composition based on zein and enriched with various natural bioactive compounds, which do not significantly deteriorate mechanical properties, while simultaneously enhancing the antimicrobial activity. Structurally variable polymer films, depending on the specific active agent, or their combinations were obtained. Although a slight decrease in tensile characteristics was observed after the zein modification, a surprisingly large increase in film elasticity was shown, especially in the films containing monolaurin with oregano oil or eugenol or the eugenol itself, where almost 97% accrual was achieved. In addition, all samples modified with active agents exhibited enhanced barrier properties; water vapor permeability values decreased by almost 80% in the zein sample with oregano oil. The wettability measurement proved increased hydrophobicity in the films containing the combinations of monolaurin with polyphenolic compounds, while slight decreases in contact angle were shown in the single-component films with oregano, thyme, and eugenol (40 to 50°). All tested samples, except zein containing thyme oil, exhibited antibacterial activity against Gram-positive *S. aureus*, whereas molds and yeasts were less sensitive. To broaden the potential applications, these promising materials could be studied using additional release tests for gas permeability and biodegradability.

It can be concluded that the obtained bioactive zein-based films with tunable surface properties represent a potential alternative to those made from non-renewable resources and are useful in packaging applications.

## Figures and Tables

**Figure 1 ijms-23-00384-f001:**
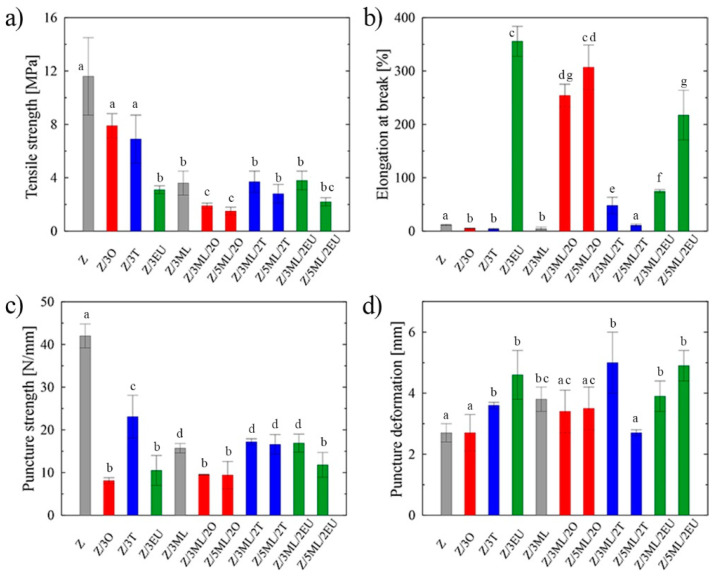
Mechanical properties of zein-based films: (**a**) tensile strength, (**b**) elongation at break, (**c**) puncture strength, (**d**) puncture deformation; Z, zein; O, oregano; T, thyme; EU, eugenol; ML, monolaurin; 2: 2 wt%; 3: 3 wt%; 5: 5 wt%. Each point represents the mean of five repetitions; error bars correspond to the standard deviation. The same letters above bars mean no significant differences among others and the different letters above bars mean significant differences (a, b, c, d, e, f, g: *p* < 0.05, ANOVA test).

**Figure 2 ijms-23-00384-f002:**
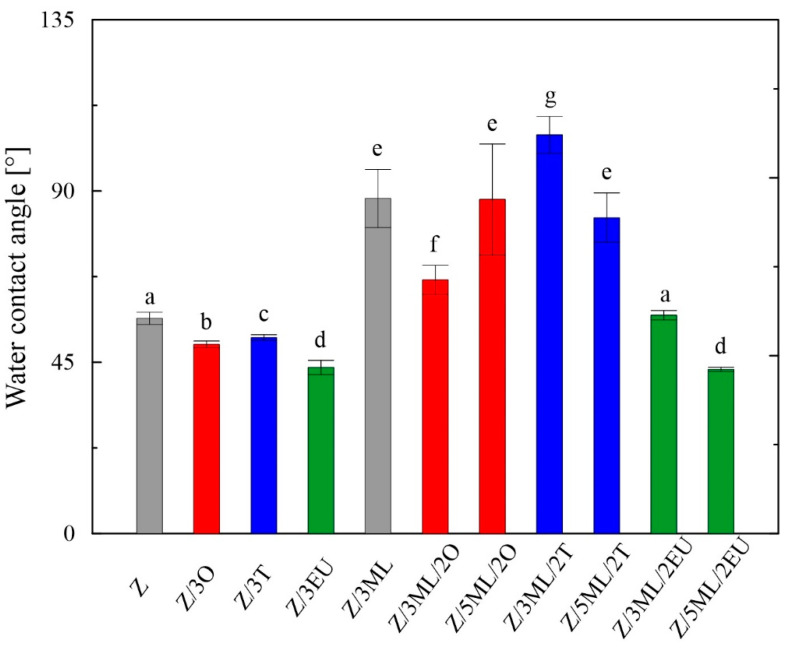
Wettability of zein-based films: Z, zein; O, oregano; T, thyme; EU, eugenol; ML, monolaurin; 2: 2 wt%; 3: 3 wt%; 5: 5 wt%. Each point represents the mean of five repetitions; error bars correspond to the standard deviation. The same letters above bars mean no significant differences among others and the different letters above bars mean significant differences (a, b, c, d, e, f, g: *p* < 0.05, ANOVA test).

**Figure 3 ijms-23-00384-f003:**
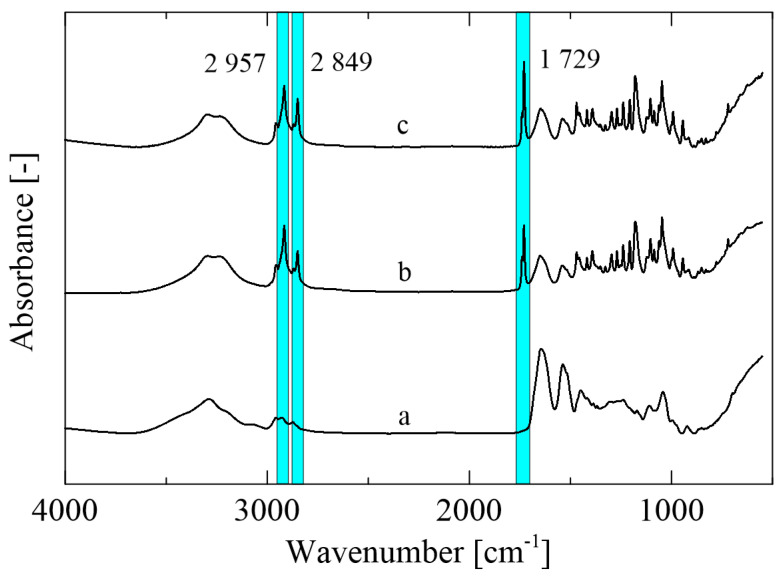
FTIR-ATR spectra of the (**a**) control zein film, (**b**) Z/3ML, and (**c**) Z/3ML/2T.

**Figure 4 ijms-23-00384-f004:**
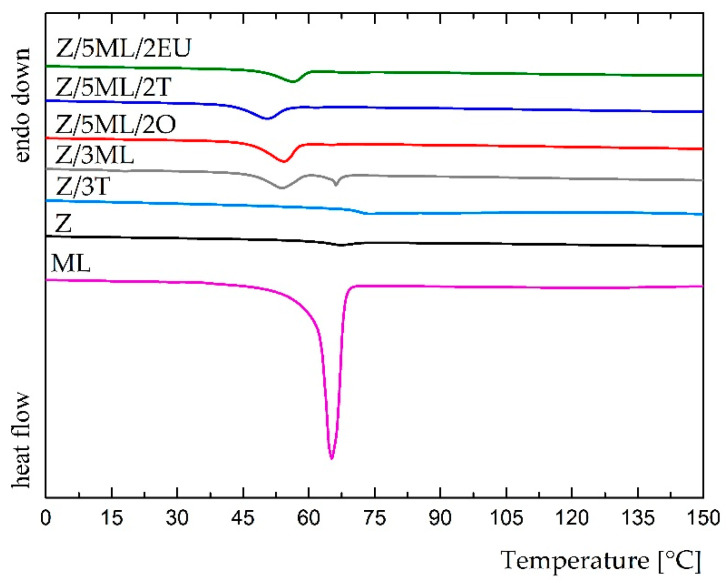
DSC thermograms of control and modified zein films (the first heating scan).

**Figure 5 ijms-23-00384-f005:**
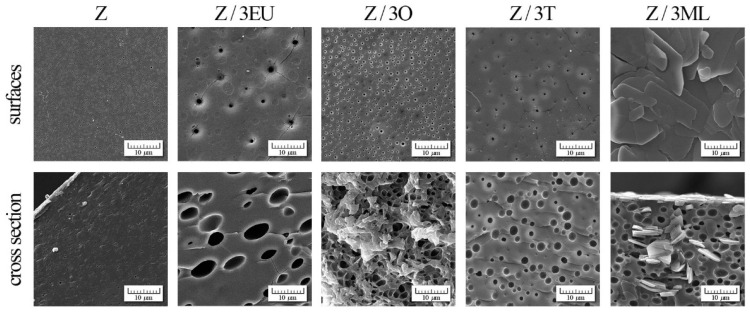
SEM photographs of zein-based films with active compounds: Z, zein; O, oregano; T, thyme; EU, eugenol; ML, monolaurin; 3: 3 wt%.

**Figure 6 ijms-23-00384-f006:**
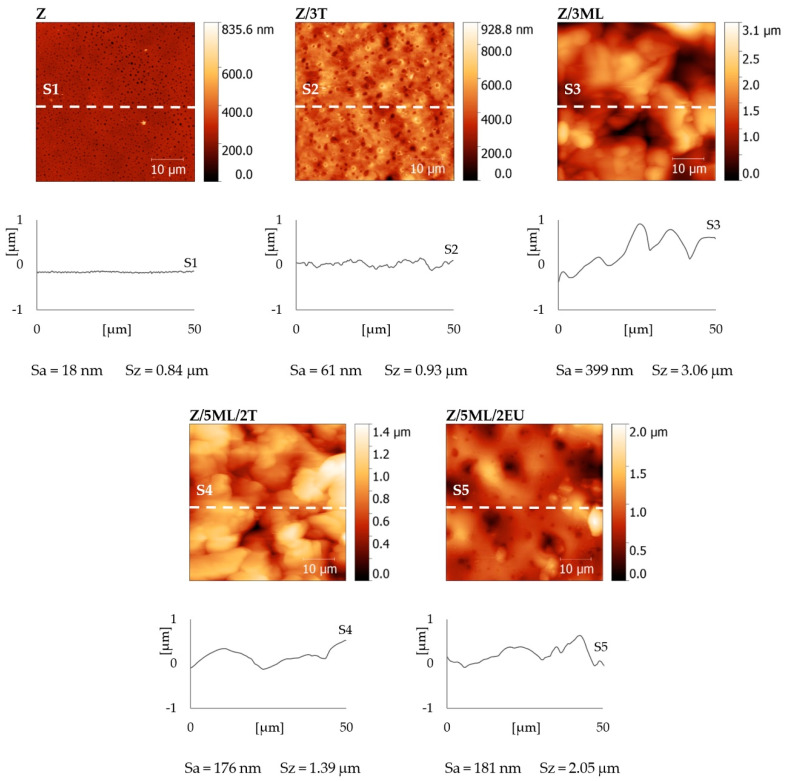
AFM height maps, surface cross-sections, and roughness (Sa) and maximal height (Sz) values of zein-based films with added active compounds. Z, zein; T, thyme; EU, eugenol; ML, monolaurin; 3: 3 wt%.

**Figure 7 ijms-23-00384-f007:**
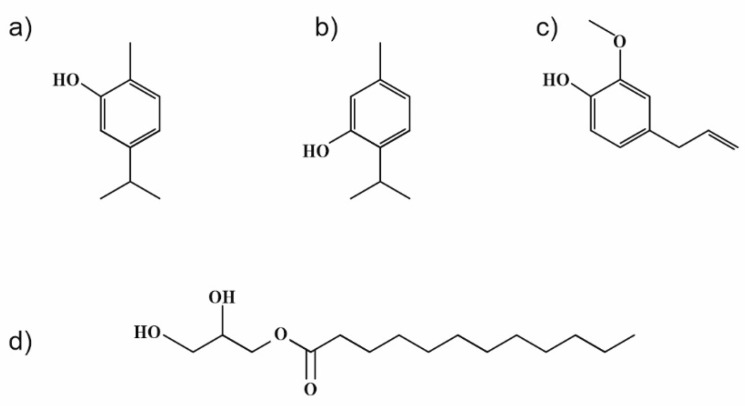
Chemical structures of (**a**) carvacrol, (**b**) thymol, (**c**) eugenol, and (**d**) 1-monolaurin.

**Figure 8 ijms-23-00384-f008:**
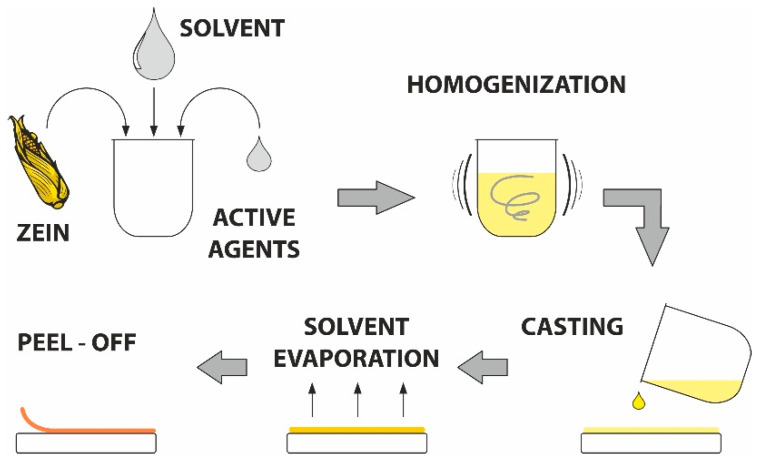
Scheme of the solvent casting process.

**Table 1 ijms-23-00384-t001:** Sample designation.

Sample	Monolaurin (wt%)	Thyme (wt%)	Oregano (wt%)	Eugenol (wt%)
Z	-	-	-	-
Z/3T	-	3	-	-
Z/3O	-	-	3	-
Z/3EU	-	-	-	3
Z/3ML	3	-	-	-
Z/3ML/2O	3	-	2	-
Z/5ML/2O	5	-	2	-
Z/3ML/2T	3	2	-	-
Z/5ML/2T	5	2	-	-
Z/3ML/2EU	3	-	-	2
Z/5ML/2EU	5	-	-	2

**Table 2 ijms-23-00384-t002:** Thickness and water vapor permeability values of zein-based films.

Sample	Thickness (µm)	WVP (g/Pa.h.m^2^)
**Z**	258 ± 4	6.14 ± 0.1
**Z/3O**	266 ± 7	1.23 ± 0.2
**Z/3T**	278 ± 16	1.37 ±0.01
**Z/3EU**	285 ± 2	1.46 ± 0.02
**Z/3ML**	174 ± 6	3.59 ± 0.07
**Z/5ML/2O**	262 ± 17	1.44 ± 0.2
**Z/5ML/2T**	175 ± 6	1.75 ± 0.03
**Z/5ML/2EU**	202 ± 14	1.82 ± 0.15

**Table 3 ijms-23-00384-t003:** Surface energy values (OWRK model).

Sample	γ_tot_ [mJ/m^2^]	γ_d_ [mJ/m^2^]	γ_p_ [mJ/m^2^]
Z	44.8 ± 1.9	9.0 ± 2.0	35.9 ± 3.8
Z/3O	57.6 ± 1.	2.1 ± 0.3	55.5 ± 1.7
Z/3T	51.9 ± 1.7	4.8 ± 1.1	47.1 ± 2.7
Z/3EU	60.5 ± 2.9	3.8 ± 0.8	56.6 ± 3.7
Z/3ML	17.3 ± 1.7	2.8 ± 0.5	14.5 ± 2.2
Z/3ML/2O	38.9 ± 2.9	4.1 ± 0.3	34.7 ± 8.3
Z/5ML/2O	18.5 ± 5.3	15.5 ± 2.5	3.0 ± 0.9
Z/3ML/2T	18.8 ± 3.9	16.8 ± 3.6	2.0 ± 0.2
Z/5ML/2T	29.3 ± 2.1	1.3 ± 0.2	28.0 ± 1.9
Z/3ML/2EU	83.0 ± 8.0	3.1 ± 1.4	79.9 ± 6.6
Z/5ML/2EU	74.0 ± 2.3	0.2 ± 0.2	73.8 ± 2.5

**Table 4 ijms-23-00384-t004:** Surface energy values (model WU).

Sample	γ_tot_ [mJ/m^2^]	γ_d_ [mJ/m^2^]	γ_p_ [mJ/m^2^]
Z	46.3 ± 1.2	12.9 ± 0.9	33.4 ± 2.1
Z/3O	53.0 ± 0.8	9.0 ± 0.2	44.0 ± 0.9
Z/3T	50.5 ± 0.8	11.0 ± 0.8	39.5 ± 1.4
Z/3EU	56.3 ± 1.6	11.6 ± 0.5	44.7 ± 2.0
Z/3ML	22.7 ± 3.4	2.6 ± 2.4	20.1 ± 9.8
Z/3ML/2O	39.8 ± 0.1	7.0 ± 5.1	32.9 ± 4.9
Z/5ML/2O	20.7 ± 1.1	13.4 ± 1.3	7.3 ± 0.2
Z/3ML/2T	19.9 ± 0.8	13.5 ± 1.8	6.5 ± 0.1
Z/5ML/2T	32.7 ± 1.6	3.4 ± 0.6	29.2 ± 1.0
Z/3ML/2EU	60.2 ± 3.6	0.7 ± 0.0	59.5 ± 4.3
Z/5ML/2EU	60.9 ± 0.9	7.0 ± 0.2	54.0 ± 1.4

**Table 5 ijms-23-00384-t005:** Antibacterial activity levels of zein-based films (9 mm disk diameter).

Sample	Inhibition zone (mm)	
	*Staphylococcus aureus*	*Escherichia coli*
Z	*	*
Z/3T	*	*
Z/3O	15 ± 0.0	*
Z/3ML	11.5 ± 0.7	*
Z/3EU	17.3 ± 0.3	14.0 ± 1.0
Z/3ML/2O	15.3 ± 0.2	*
Z/5ML/2O	21 ± 0.2	13.5 ± 0.7
Z/3ML/2T	18 ± 0.9	*
Z/5ML/2T	18.4 ± 0.6	*
Z/3ML/2EU	22.3 ± 0.3	10.0 ± 0.0
Z/5ML/2EU	20.7 ± 0.7	12 ± 0.2

* No inhibition zone, but no growth under the disk.

**Table 6 ijms-23-00384-t006:** Antifungal activity levels of zein-based films (9 mm disk diameter).

Sample	Inhibition zone (mm)	
	*Candida albicans*	*Aspergillus niger*
Z	*	*
Z/3T	*	*
Z/3O	*	*
Z/3ML	*	*
Z/3EU	11.3 ± 0.5	*
Z/3ML/2O	*	*
Z/5ML/2O	16.7 ± 0.8	13.1 ± 0.0
Z/3ML/2T	*	*
Z/5ML/2T	*	*
Z/3ML/2EU	17.2 ± 0.5	15.9± 0.8
Z/5ML/2EU	15.0 ± 0.1	11.3 ± 0.3

* No inhibition zone, but no growth under the disk.

**Table 7 ijms-23-00384-t007:** Modified Gompertz equation fitting results for *E. coli* and *S. aureus* growth treated by zein and zein/oregano (O), monolaurin (ML), and eugenol (EU). Note: λ is lag phase (h); A is asymptote defined as the maximum theoretical achieved value of relative number of microorganisms (Log CFU.L^−1^); μ_max_ is the maximum specific growth rate (Log CFU.L^−1^.h^−1^); Adj. R^2^ is theregression coefficient.

Bacteria	Sample	λ (h)	A (Log CFU.L^−1^)	μ_max_ (Log CFU.L^−1^.h^−1^)	Adj. R^2^
*E. coli*	Control	4.146 ± 0.038 ^a^	0.483 ± 0.029 ^a^	0.602 ± 0.018 ^a^	0.997
	Z	3.792 ± 0.034 ^a^	0.353 ± 0.036 ^a^	0.525 ± 0.012 ^a^	0.998
	Z/3O	-*	-	-	-
	Z/5ML/2O	-	-	-	-
	Z/3EU	-	-	-	-
	Z/5ML/2EU	-	-	-	-
*S. aureus*	Control	6.217 ± 0.201 ^a^	0.402 ± 0.008 ^a^	0.269 ± 0.022 ^a^	0.999
	Z	6.759 ± 0.144 ^a^	0.428 ± 0.016 ^a^	0.231± 0.012 ^a^	0.988
	Z/3O	-	-	-	-
	Z/5ML/2O	-	-	-	-
	Z/3EU	-	-	-	-
	Z/5ML/2EU	-	-	-	-


^a^ Different upper case letters in the same column for each bacteria indicate significant differences (*p* < 0.05) (ANOVA); * -, no growth.

## Data Availability

Data are contained within the article and Appendix A.

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
