# Peer review of "Zein-Based Films Containing Monolaurin/Eugenol or Essential Oils with Potential for Bioactive Packaging Application"

_ijms, 2021, doi:10.3390/ijms23010384_

Round 1

Reviewer 1 Report

Dear authors!

My comments  are in attached file.

Author Response

Response to Reviewer 1

Dear reviewer,

we are grateful for your insightful comments on our manuscript. We suppose that we incorporated changes covering your suggestions. All the revisions were highlighted in manuscript using the “track Changes” in Microsoft Word. The answers to the individual comments follow.

Thank you for your re-consideration of this manuscript.

Sincerely,

Jana Sedlarikova

The article Sedlarikova et al. “Zein-based Films as Carriers of Bioactive Monolaurin/phenolic Compounds” is devoted to the characteristics of the zein-based films enriched with bioactive organic compounds. The MS contains some interesting data, which can be applied in practice, however, I have some critical comments:

1) The DISCUSSION section is practically absent. It needs supplementing with the latest data on the research topic. Perhaps, it makes sense separating the DISCUSSION into a separate section. In my opinion, it will be more informative for the readers.

The Discussion was extended and placed into separate section. Some other analyses (DSC, AFM) of the samples were included to improve the interpretation of obtained results.

2) The present figures are of poor resolution, possibly in my version of the manuscript. Their quality must be improved.

Thank you for pointing this out. The quality of Figures was probably lowered due to converting into pdf format. Figures were downloaded in a separate file and quality and resolution was verified.

3) In addition, in Figures 1 and 2, statistical analysis is not presented within the framework of ANOVA statistics (the comparison of experimental variants with one another), as well as in the caption to the figure.

Thank you for the comment. ANOVA statistics was added in the given Figures.

Reviewer 2 Report

The article “Zein-based Films as Carriers of Bioactive Monolaurin/phenolic Compounds” presents the results of the preparation and characterization of biodegradable zein-based films for potential application as the active packaging materials. The subject is very important and biodegradable films are very desirable, hence the manuscript on this subject is scientifically significant. I recommended to publish this work after major revisions. To this purpose, I suggest the following actions:

1) Preparation of film: In order to better illustrate the preparation method, I propose to draw a scheme of obtaining the films and include photos of them.

2) Results:

  • Please, add the results of thermal properties of investigated films (TGA, DSC)
  • The information provided in the article is not sufficient to assess whether the investigated biodegradable films are suitable for food purposes. To determine this, please carry out the measure of the oxygen and carbon dioxide permeability through the obtained films.
  • In general, the description of the obtained results is very general and not very scientific. I propose to delve deeper into the presented topic and explain more scientifically why some additives are better than others.

3) Conclusion: The conclusions are too general. Please add more detailed information and a few values to enrich the conclusions. The authors should also give a future recommendations to the new authors in the field.

Author Response

Response to Reviewer 2

Dear reviewer,

we are grateful for your insightful comments on our manuscript. We suppose that we incorporated changes covering your suggestions. All the revisions were highlighted in manuscript using the “track Changes” in Microsoft Word. The answers to the individual comments follow.

Thank you for your re-consideration of this manuscript.

Sincerely,

Jana Sedlarikova

The article “Zein-based Films as Carriers of Bioactive Monolaurin/phenolic Compounds” presents the results of the preparation and characterization of biodegradable zein-based films for potential application as the active packaging materials. The subject is very important and biodegradable films are very desirable, hence the manuscript on this subject is scientifically significant. I recommended to publish this work after major revisions. To this purpose, I suggest the following actions:

1) Preparation of film: In order to better illustrate the preparation method, I propose to draw a scheme of obtaining the films and include photos of them.

The scheme of the solvent casting proces was added into the manuscript and the photos of the prepared films were included in Supplementary Material (Figure S3).

2) Results:

  • Please, add the results of thermal properties of investigated films (TGA, DSC)
  • Authors thank for the comment. DSC measurement was carried out and discussed in an appropriate section.
  • The information provided in the article is not sufficient to assess whether the investigated biodegradable films are suitable for food purposes. To determine this, please carry out the measure of the oxygen and carbon dioxide permeability through the obtained films.
  • Thank you for this valuable comment. Unfortunately we are not able to fulfil this requirement because of the non-availability of these tests at our Department. For this reason, these would have to be carried out externally, which is not possible in the given time. Authors plan to do suggested experiments within further work on selected optimized formulations of zein based films, which was the objective of submitted manuscript.
  • In general, the description of the obtained results is very general and not very scientific. I propose to delve deeper into the presented topic and explain more scientifically why some additives are better than others.
  • The description of results and the discussion was extended including the more explanations of potential interactions between the individual components.

3) Conclusion: The conclusions are too general. Please add more detailed information and a few values to enrich the conclusions. The authors should also give a future recommendations to the new authors in the field.

Thank you for the comment. Conclusion was rewritten to be more specific and include more detailed information.

Reviewer 3 Report

Manuscript ijms-1511233 presents the physicochemical and antimicrobial properties of the zein-based films enriched with bioactive organic compounds. The manuscript needs significant improvements before being considered for publication.

The title needs to be modified to reflect the manuscript content better. The used essential oil (oregano and thyme oil) consists of phenolic compounds (primarily carvacrol and thymol, respectively, thymol). The eugenol has a polyphenol structure. However, the most important characteristic of these bioactive compounds is the (semi)volatility; therefore, I suggest replacing polyphenol with ”polyphenolic essential oils.”

The abstract must be re-written. I suggest using the IJMS template for Abstract. A part of the content of the Conclusion Section could be moved here.

The introduction section must include a presentation of the aim and scope of the work and briefly mention the main conclusion of the work.

Figures 1 and 2 include error bars. Figures caption did not disclose the type of error bars (Standard Error of the Mean or Standard Deviation) and the number of repetitions.

Figures 3 and Figures 4 are low-quality figures. Figure 4 is challenging to read.

The surface energy calculation does not consider the topography of the zein film surface. This is a weakness of the manuscript. My suggestion is to measure the surface topography at the nanoscale by atomic force microscopy. Or at least please discuss the influence of topography on surface energy. Please read - Malm, M. J., Narsimhan, G., & Kokini, J. L. (2019). Effect of the contact surface, plasticized and crosslinked zein films are cast on dispersive and polar surface energy distribution using the Van Oss method of deconvolution. Journal of Food Engineering, 263, 262-271.  

Please use the correct form for the scientific name of the microorganisms, i.e., Italics.

In the Material and Method section, zein characteristics must be presented (at least protein content and producer). Significant components of oregano (carvacrol and thymol) and thyme (thymol) essential oil must be shown.

The conclusion Section is too long. My suggestion is to reduce it to the main finding of the work.

Author Response

Response to Reviewer 3

Dear reviewer,

we are grateful for your insightful comments on our manuscript. We suppose that we incorporated changes covering your suggestions. All the revisions were highlighted in manuscript using the “track Changes” in Microsoft Word. The answers to the individual comments follow.

Thank you for your re-consideration of this manuscript.

Sincerely,

Jana Sedlarikova

Manuscript ijms-1511233 presents the physicochemical and antimicrobial properties of the zein-based films enriched with bioactive organic compounds. The manuscript needs significant improvements before being considered for publication.

The title needs to be modified to reflect the manuscript content better. The used essential oil (oregano and thyme oil) consists of phenolic compounds (primarily carvacrol and thymol, respectively, thymol). The eugenol has a polyphenol structure. However, the most important characteristic of these bioactive compounds is the (semi)volatility; therefore, I suggest replacing polyphenol with ”polyphenolic essential oils.”

Thank you for the valuable comments. The title was modified to correspond more to the content of manuscript.

The abstract must be re-written. I suggest using the IJMS template for Abstract. A part of the content of the Conclusion Section could be moved here.

Authors thank for the comment. Abstract was rewritten according to the suggested IJMS template.

The introduction section must include a presentation of the aim and scope of the work and briefly mention the main conclusion of the work.

The aim, scope, as well as the main conclusion were included in the Introduction section.

Figures 1 and 2 include error bars. Figures caption did not disclose the type of error bars (Standard Error of the Mean or Standard Deviation) and the number of repetitions.

Thank you for the comment, the error bars correspond to Standard deviation. It was added into appropriate Figure captions including the number of repetitions.

Figures 3 and Figures 4 are low-quality figures. Figure 4 is challenging to read.

Thank you for pointing this out. The quality of Figures was probably lowered due to converting into pdf format. Figures were downloaded in separate file and quality and resolution was verified.

The surface energy calculation does not consider the topography of the zein film surface. This is a weakness of the manuscript. My suggestion is to measure the surface topography at the nanoscale by atomic force microscopy. Or at least please discuss the influence of topography on surface energy. Please read - Malm, M. J., Narsimhan, G., & Kokini, J. L. (2019). Effect of the contact surface, plasticized and crosslinked zein films are cast on dispersive and polar surface energy distribution using the Van Oss method of deconvolution. Journal of Food Engineering, 263, 262-271. 

Authors thank for this relevant comment and valuable publication to discuss. The AFM analysis was carried out and added in an appropriate section (2.8).

Please use the correct form for the scientific name of the microorganisms, i.e., Italics.

Thank you for the comment, the mistake in letter font probably occurred during the file conversion into pdf format. It was corrected in the resubmitted version.

In the Material and Method section, zein characteristics must be presented (at least protein content and producer). Significant components of oregano (carvacrol and thymol) and thyme (thymol) essential oil must be shown.

Authors apologize for the absence of polymer characterization. It was added in the Materials and Methods section, together with chemical structures of carvacrol and thymol, as the main constituents of used oregano and thyme essential oils, respectively, as well as eugenol and 1-monolaurin structure was included (Figure 7).

The conclusion Section is too long. My suggestion is to reduce it to the main finding of the work.

Thank you for the comment, conclusion was rewritten to be more specific as regards the main findings of the work.

Round 2

Reviewer 1 Report

Dear Authors!

my comments are in attached file

Author Response

Dear reviewer,

we are grateful for your comment. We made the changes in the manuscript to cover your suggestion. The revisions were highlighted in manuscript using the “track Changes” in Microsoft Word. The answer to the comment follows.

Thank you for your re-consideration of this manuscript.

Sincerely,

Jana Sedlarikova

The manuscript has been substantially revised. Unfortunately, I have to leave some critical comments:

In Figure 1, the authors indicated the groups for comparison. However, it is not clear with what, for example, options e and f (panel b) and a (panel c) are compared? You also need to clarify with which option the presented data are compared (with Z?). There is no explanation in the caption to the figure or in the text. In the Fig 2 the same goes for options b, g and f. Maybe, for clarity, indicate those variants that are significantly different?

The authors are grateful for the comment. The results were compared to each other by ANOVA test, not only versus control zein film (Z). For all variables with the same letter, the difference between the means is not statistically significant. If two variables have different letters, they are significantly different at a selected level of significance (p < 0.05). In the case of our samples, the addition of different active compounds, or their combinations, led to quite significant results, especially in mechanical and surface properties, which led to the different superscript letters in the discussed graphs. E.g., in Figure 1 panel b – Z is the same only as Z/ML/2T; all others are statistically different to the Z; next Z/3O is the same as Z/3T and Z/3ML; Z/3EU is the same as Z/5ML/2O and different to all others; Z/3ML/2O is the same as Z/5ML/2O and different to all others; Z/3ML/2T is different to all others; Z/3ML/2EU is different to all others; Z/5ML/2EU is different to all others.   

The explanation was extended in the manuscript in appropriate Figures captions (Figure 1 and 2).

Figure 1. Mechanical properties of zein-based films: Z, zein; O, oregano; T, thyme; EU, eugenol; ML, monolaurin; 2: 2 wt%; 3: 3 wt%; 5: 5 wt%. Each point represents the mean of five repetitions; error bars correspond to the standard deviation. The same letters above bars mean no significant differences among others and the different letters above bars mean significant differences (a, b, c, d, e, f: p < 0.05, ANOVA test).   

Figure 2. Wettability of zein-based films: Z, zein; O, oregano; T, thyme; EU, eugenol; ML, monolaurin; 2: 2 wt%; 3: 3 wt%; 5: 5 wt%. Each point represents the mean of five repetitions; error bars correspond to the standard deviation. The same letters above bars mean no significant differences among others and the different letters above bars mean significant differences (a, b, c, d, e, f: p < 0.05, ANOVA test).   

Reviewer 3 Report

Authors made the requested imrpovements and manuscript could be published.

Author Response

Dear reviewer,

Thank you very much for the kind reconsideration of our manuscript and acceptance of the changes we made.

Sincerely,

Jana Sedlarikova